# LSTM-based estimation of lithium-ion battery SOH using data characteristics and spatio-temporal attention

Gengchen Xu[1], Jingyun Xu[1,2,3]*, Yifan Zhu[1]

**1** School of Engineering, Huzhou University, Huzhou, P. R. China, **2** Huzhou Key Laboratory of Intelligent Sensing and Optimal Control for Industrial Systems, School of Engineering, Huzhou University, Huzhou, P. R. China, **3** Huzhou Key Laboratory of Green Energy Materials and Battery Cascade Utilization, School of Intelligent Manufacturing, Huzhou College, Huzhou, P. R. China

* 00605@zjhu.edu.cn

**Data Availability Statement:** This dataset is provided by the NASA Prognostics Center of Excellence (PCoE) as part of the NASA Lithium-Ion Battery Data Set (https://www.nasa.gov/intelligent-systems-division/discovery-and-systems-health/

## Abstract

As the primary power source for electric vehicles, the accurate estimation of the State of Health (SOH) of lithium-ion batteries is crucial for ensuring the reliable operation of the power system. Long Short-Term Memory (LSTM), a special type of recurrent neural network, achieves sequence information estimation through a gating mechanism. However, traditional LSTM-based SOH estimation methods do not account for the fact that the degradation sequence of battery SOH exhibits trend-like nonlinearity and significant dynamic variations between samples. Therefore, this paper proposes an LSTM-based lithium-ion SOH estimation method incorporating data characteristics and spatio-temporal attention. First, considering the trend-like nonlinearity of the degradation sequence, which is initially gradual and then rapid, input features are filtered and divided into trend and non-trend features. Then, to address the significant dynamic variations between samples, especially for capacity regeneration, a spatio-temporal attention mechanism is designed to extract spatio-temporal features from multidimensional non-trend features. Subsequently, an LSTM model is built with trend features, spatio-temporal features, and actual capacity as inputs to estimate capacity. Finally, the model is trained and tested on different datasets. Experimental results demonstrate that the proposed method outperforms traditional methods in terms of SOH estimation accuracy and robustness.

## Introduction

Lithium-ion batteries, known for their high energy density, high conversion efficiency, and absence of memory effect, are widely used in electric vehicles and energy storage systems [1]. Due to repeated charging and discharging, internal aging of the battery occurs, including degradation of anode and cathode materials, aging of the separator, and reduction of the electrolyte, which leads to decreased battery capacity, increased internal resistance, and elevated self-discharge. The State of Health (SOH) of a lithium-ion battery typically refers to the ratio of its capacity to its nominal capacity under specific discharge conditions. The process involves

pcoe/pcoe-data-set-repository/). The dataset contains experimental data on lithium-ion batteries, including charge and discharge cycles at different temperatures, with recorded impedance as the criterion for damage. This dataset has been widely used in the research of battery health management and prognostics.Details are as follows: Download link: https://phm-datasets.s3.amazonaws.com/NASA/5.+Battery+Data+Set.zipData set citation: B. Saha and K. Goebel (2007). "Battery Data Set," NASA Prognostics Data Repository, NASA Ames Research Center, Moffett Field, CA. Regarding the Health Factor Dataset, this dataset was generated through experimental procedures and primarily reflects the battery health under different voltage ranges. These health factors are derived from the battery's capacity changes across different voltage ranges and specific stages, providing an accurate reflection of the overall battery health status. If you or any reader requires this dataset, Please send an email to 1978083595@qq.com to contact (Qian Zhuge) for further details.

**Funding:** This research was partially supported by Zhejiang Provincial Natural Science Foundation of China under Grant (No. LTGS23E070002).

**Competing interests:** The authors have declared that no competing interests exist.

charging the battery to full capacity [2], followed by discharging it at a defined rate until the cutoff voltage is reached. Accurate SOH estimation for lithium-ion batteries can extend battery life, enhance safety, and ensure timely battery replacement, thus maintaining efficient and stable operation of the battery [3,4].

Current mainstream methods for estimating the State of Health (SOH) of lithium-ion batteries can be broadly classified into two categories: model-based methods and data-driven methods [5]. Model-based methods are constructed based on the electrochemical evolution laws within the battery, taking full account of the influence of various aging factors on internal and external state variables, and thus developing battery aging models [6]. The main approaches include Kalman filtering, particle filtering, electrochemical impedance spectroscopy, and equivalent circuit models, among others [7]. Wang et al. developed a state-space model for lithium-ion battery capacity degradation using a cubature particle filter and conducted SOH estimation [8]. Sun et al. proposed a battery SOH estimation method based on a simplified electrochemical model optimized using a back-propagation neural network [9]. Zhang et al. combined solid electrolyte interface resistance and charge transfer resistance with temperature and State of Charge (SOC) to establish a probabilistic model for SOH estimation [10]. Liu et al. developed a lithium-ion battery dynamic model based on the open-circuit voltage method and further refined the model by integrating internal resistance correction and Kalman filtering for parameter adjustment [11]. Through effective use of state-of-charge estimation, the accuracy of capacity estimation was maintained at a high level. However, model-based methods need to consider the internal physicochemical properties of the battery, making the modeling process complex and challenging. Additionally, these methods exhibit poor generalization when applied to complex usage environments and varying operating conditions [12].

Data-driven methods do not require consideration of the internal electrochemical reactions and failure mechanisms of lithium-ion batteries. Instead, they derive insights from battery performance test data and state monitoring data, extracting the latent health status information and its evolutionary trends, thus enabling SOH estimation [13]. These methods can, to some extent, overcome the challenges of modeling difficulties and poor generalization faced by model-based methods [14]. The main approaches include support vector machines, autoregressive models, Gaussian process regression models, and Long Short-Term Memory (LSTM) networks. Lin et al. utilized the correlations between multimodal multilinear features and designed a high-order polynomial module to integrate feature information, thereby improving the efficiency and performance of SOH estimation [15]. Wang et al. developed a support vector regression model to predict the remaining useful life of lithium-ion batteries and optimized the penalty factor and kernel parameters using the artificial bee colony algorithm for SOH estimation [16]. Chen et al. employed empirical mode decomposition to decompose capacity degradation data and built autoregressive moving average (ARMA) and Elman neural network models to train and predict multiple modal sequences and residuals separately [17]. Yun et al. selected three temporal health indicators from the battery's charge-discharge data and applied the naive Bayes Monte Carlo theory to estimate the SOH of the battery [18]. Ren et al. proposed an SOH estimation method that combines convolutional neural networks and LSTM networks, demonstrating its effectiveness on real-world datasets [19]. Liu et al. decomposed battery capacity data into intrinsic mode functions and residuals using empirical mode decomposition and employed Gaussian process regression and LSTM networks to estimate the intrinsic mode functions and residuals, respectively. This approach directly captures the long-term dependency relationships of capacity, enabling accurate predictions of capacity and SOH [20].

By comparing the aforementioned methods, it is evident that LSTM has stronger sequence modeling and nonlinear learning capabilities in SOH estimation for lithium-ion batteries compared to support vector machines, autoregressive models, and Gaussian process regression models [21]. However, LSTM still faces the challenge of being unable to focus on different variables at different time steps. To address this limitation, Yuan et al. proposed a Spatio-temporal Attention-based LSTM model (STA-LSTM) for industrial process quality forecasting [22]. The model can identify samples and input variables related to quality variables at different time steps within the time window, thereby improving prediction accuracy.However, the SOH degradation sequence of lithium-ion batteries inherently exhibits trend-like nonlinearity and significant dynamic variations between samples. There are two key issues with applying the STA-LSTM model for SOH estimation: 1) The SOH degradation sequence shows a trend that is initially gradual and then rapidly declines. In SOH samples, there are variables that reflect long-term trends (trend variables) and others that reflect short-term changes (short-term variables) [23]. The STA-LSTM model applies spatial attention weighting to all sample variables and propagates them upwards through layers, which obscures the influence of trend variables on long-term trends. 2) The model's temporal attention mechanism adaptively identifies the weights of different sample features relevant to quality prediction but determines these weights based on estimated quality values rather than actual quality values [24]. This lack of actual quality data in calculating the weights significantly affects the estimation accuracy for SOH sequences with pronounced dynamic variations. To address these issues, this paper proposes an LSTM-based SOH estimation method for lithium-ion batteries that incorporates data characteristics and spatio-temporal attention. First, to address the trend-like nonlinearity of the degradation sequence, which starts off gradually and then accelerates, input features are filtered and divided into trend and non-trend features. Then, to account for the significant dynamic variations between samples, particularly the issue of capacity regeneration, a spatio-temporal attention mechanism structure is designed to extract spatio-temporal features from multidimensional non-trend features. Subsequently, an LSTM model is built using trend features, spatio-temporal features, and actual capacity as inputs to estimate battery capacity. The main contributions of this paper are as follows:

1. To address the trend-like nonlinearity in the degradation sequence, which is initially gradual and then accelerates, the input variables are filtered based on the degradation characteristics of lithium-ion batteries and classified into trend variables and short-term variables. Trend variables represent the long-term trends of SOH degradation, while short-term variables capture short-term changes.

2. To address the significant dynamic variations between samples, particularly the issue of capacity regeneration, a spatio-temporal attention mechanism for multidimensional short-term variables is designed. This mechanism reduces redundancy between different features through spatial attention and uses temporal attention to steadily predict the stable parts of capacity degradation while effectively tracking the sudden rises caused by capacity regeneration.

3. An LSTM structure based on variable classification and spatio-temporal attention is developed to estimate SOH.

The remainder of this paper is organized as follows: Section 2 introduces the battery training and degradation datasets and analyzes the correlation between the selected health factors and SOH. Section 3 provides a detailed description of the fusion model based on data characteristics and the spatio-temporal attention mechanism. Section 4 presents the experimental results and analysis. Finally, Section 5 concludes the paper.

## Battery training and degradation datasets

To study the aging characteristics of lithium-ion batteries, NASA's 18650-type lithium-ion battery was used, with a rated capacity of 2Ah and a rated voltage of 3.6V. The charge and discharge cutoff voltages were set to 4.2V and 2.5V, respectively. All data in this paper were collected at ambient temperature, using constant current (CC) and constant voltage (CV) for charging, and constant current (CC) for discharging. Specifically, during the charging process, the battery was charged in CC mode at a constant current of 1C until reaching 4.2V, followed by CV mode charging until the charging current dropped to 20mA [25]. To simulate different operating conditions in an ideal laboratory environment, the discharge process was conducted in CC mode, designed with five different discharge rates and intervals, discharging to the cutoff voltage of 2.5V. The detailed voltage and current variation curves for battery charge and discharge are shown in Fig 1. The aging test data of the battery were used to evaluate the performance of the developed SOH and Remaining useful life (RUL) prediction methods in practical applications. Therefore, this paper selected multiple health features as inputs to the model, extracted from the voltage data obtained during the charging process, which best reflect the battery's operational characteristics. Fig 2 shows the charging voltage variation curves of the B5 battery from the NASA dataset under different cycle numbers. From the figure, it can be clearly observed that as the number of charge-discharge cycles increases, the degree of battery aging intensifies, and the duration of the CC charging phase gradually decreases [26]. During the CC charging process, the time duration between different voltage intervals also decreases as the cycle number increases. This paper defines this duration as the time spent in different voltage ranges during charging, and uses it as a health feature.

This paper selects three sets of data, labeled B5, B6, and B7, as the subjects for health factor extraction. Under the CC charging mode, the time required for the battery voltage to move through four stages: 3.8V-3.9V (F1), 3.9V-4.0V (F2), 4.0V-4.1V (F3), and 4.1V-4.2V (F4) is recorded [27]. Among them,F1 represents the battery performance in the 3.8V-3.9V range, primarily reflecting capacity changes in the initial phase; F2 represents the 3.9V-4.0V range, reflecting the battery's mid-term health state; F3 represents the 4.0V-4.1V range, tracking the

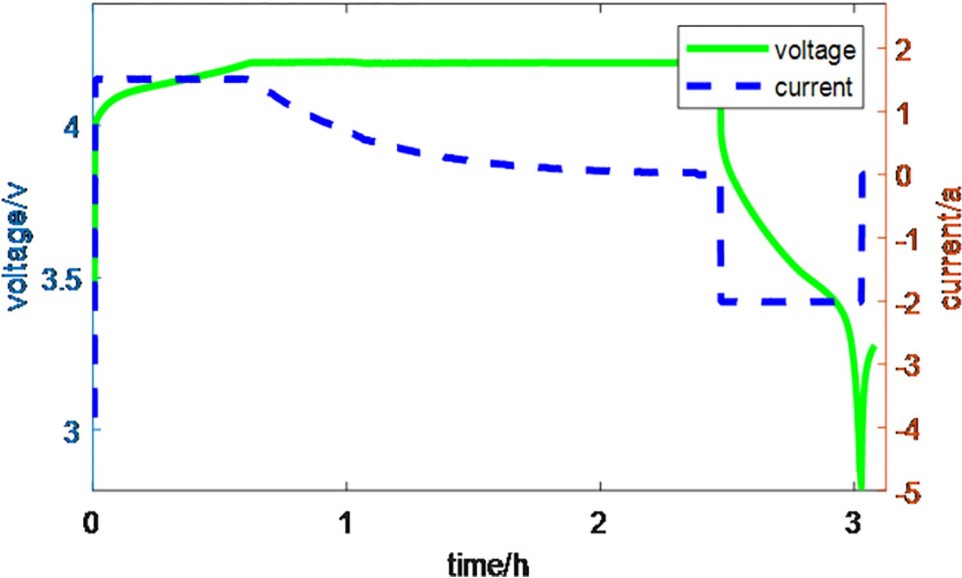

**Fig 1. Curve of battery charging and discharging voltage and current.**

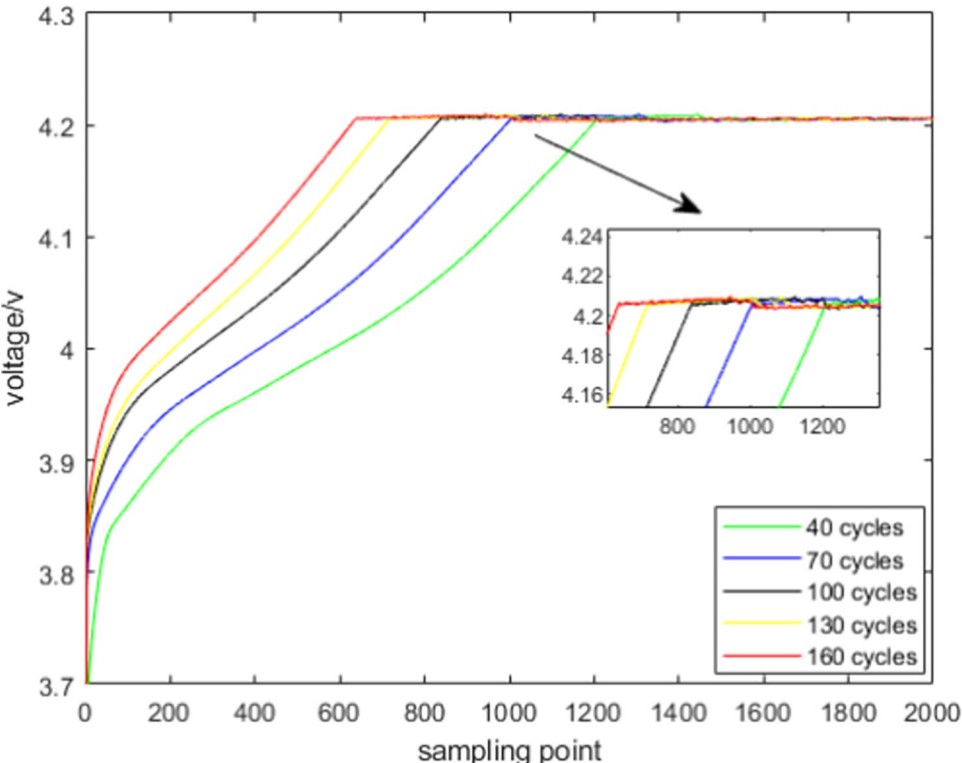

**Fig 2. Charging voltage variation curve of B5 under different cycles.**

battery's performance as it approaches full charge; F4 represents the 4.1V-4.2V range, describing the battery's behavior under high voltage conditions.The experimental results are shown in Fig 3, illustrating the changes in capacity and health factors over the number of cycles. From the figure, the following observations can be made: (1) As the number of cycles increases, both the capacity and the four health factors gradually decrease overall; (2) When capacity regeneration occurs at certain cycle numbers, the capacity increases, and the four health factors also increase to varying degrees; (3) Among the four health factors, F1, F2, and F3 are more closely related to capacity overall, while F4 shows higher correlation with capacity in the later stages.

To further refine the selection of health factors, this paper calculates the Spearman correlation coefficient between the capacity and the four health factors. The calculation formula is as follows:

$$r_s = \frac{\sum_{k=1}^{N}(x_k - \bar{x})(y_k - \bar{y})}{\sqrt{\sum_{k=1}^{N}(x_k - \bar{x})^2 \sum_{k=1}^{N}(y_k - \bar{y})^2}} \tag{1}$$

Where $r_s$ represents the Spearman correlation coefficient, $x_k$ and $y_k$ denote the health factor and the cycle capacity at the k-th measurement, respectively, and $\bar{x}$ and $\bar{y}$ represent the average values of the N health factors and cycle capacities, with N being the number of elements. The statistical results are shown in Table 1, which presents the correlation between capacity and health factors. From the table, the following conclusions can be drawn: (1) The correlation coefficients between the health factors F1, F2, and F3 and capacity are all greater than 0.9,

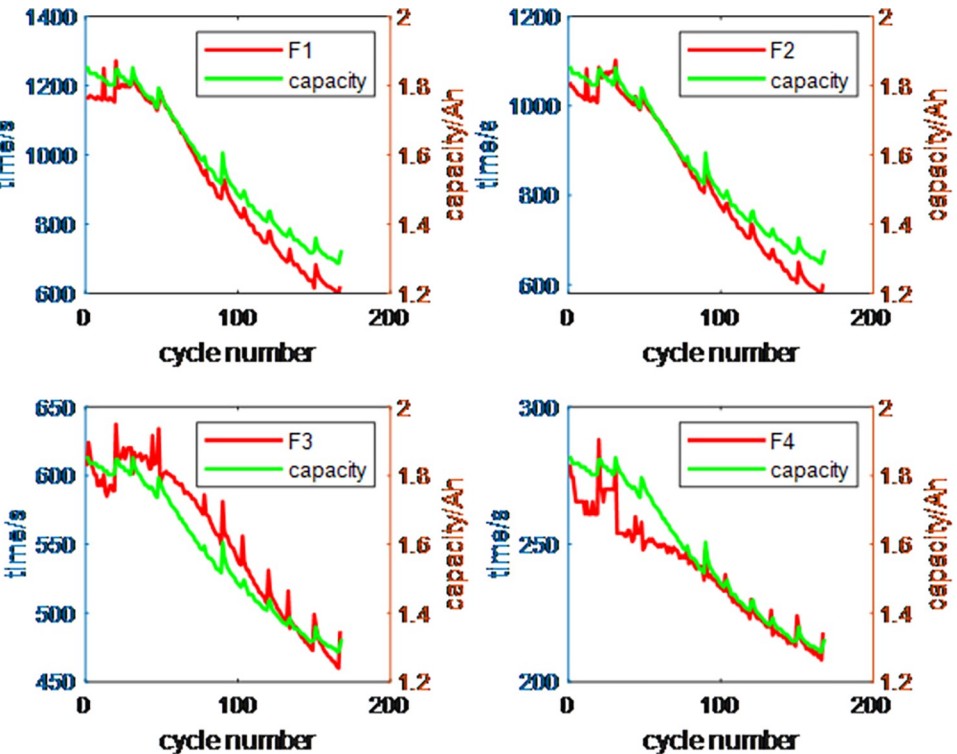

**Fig 3. Variation of volume and health factors with the number of cycles.**

indicating a strong correlation with capacity; (2) For batteries B5 and B7, the health factor F1 shows the highest correlation with capacity, while for battery B6, F2 has the highest correlation; (3) For data with cycle numbers greater than 90, F4 exhibits the highest correlation. The data sets F1, F2, and F3 show redundancy. Since users typically charge the battery when there is still remaining capacity, the health factor F1 may have missing data. Subsequent experimental validation shows that F2 and F3 perform best as health factors, and therefore, F2 and F3 are selected for further research in this paper.

## LSTM model based on the fusion of data characteristics and spatio-temporal attention mechanism

To address the trend-like nonlinearity of degradation sequences, which is initially gradual and then accelerates, the degradation data is divided into trend data and short-term data. Then, to handle the significant dynamic variations between samples, particularly in the case of capacity regeneration, a spatio-temporal attention mechanism is designed for short-term data to extract short-term spatio-temporal features [28]. An LSTM model is then constructed using trend data, short-term spatio-temporal features, estimated capacity, and actual capacity as inputs to estimate future capacity. The overall structure of the algorithm is shown in Fig 4. Block I is a

**Table 1. Correlation between capacity and health factors.**

| Battery | 3.8V-3.9V | 3.9V-4.0V | 4.0V-4.1V | 4.1V-4.2V |
|---|---|---|---|---|
| B0005 | 0.993 | 0.992 | 0.993 | 0.989 |
| B0006 | 0.989 | 0.990 | 0.988 | 0.394 |
| B0007 | 0.978 | 0.975 | 0.976 | 0.985 |

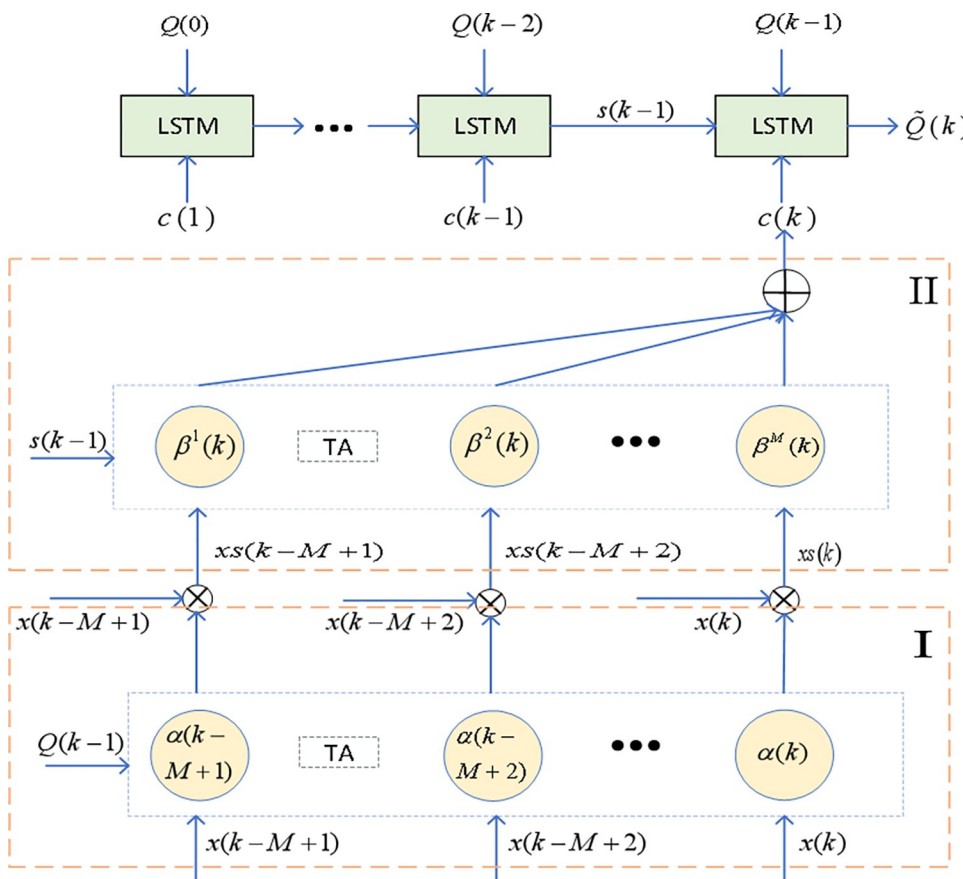

**Fig 4. Algorithm structure diagram.**

feature optimization module based on the spatial attention mechanism. This module mainly addresses the issue of feature selection by combining the health factors F2, F3, and $Q$ to form a three-dimensional vector, and using the attention mechanism to calculate weights, enabling the model to better capture the patterns of battery capacity changes over time.Block II is a feature optimization module based on the temporal attention mechanism. This module focuses on dynamic changes in the time series, paying special attention to key time points. It effectively addresses prediction errors caused by fluctuations in time features during battery capacity prediction, further improving the model's stability and prediction accuracy.

### Short-term feature extraction

To capture the long-term dependencies in battery capacity degradation data, LSTM units are used as the basic activation function units in the proposed STL-LSTM model. In LSTM, three gate controllers and a memory cell are embedded within each basic LSTM unit, namely the input gate, forget gate, and output gate. These three gates are used to determine which information from the weighted time series should be remembered. The memory cell stores the input information across all time steps. The LSTM network achieves temporal memory through the switching of these gates and the memory cell, preventing the vanishing gradient problem [29]. The structure of the short-term feature extraction process is shown in Fig 5, where the input variable $c(k) = [c_1(k), c_2(k), \cdots, c_T(k)]$ represents the short-term data at the $k$-th cycle, and $T$ denotes the dimension of the short-term data. The output variable represents

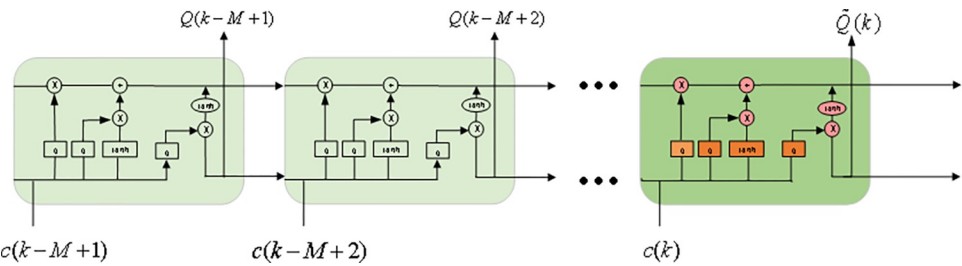

**Fig 5. Short-time feature extraction structure.**

the hidden state at the $k$-th cycle. The input variables $c(k)$ is processed through the LSTM unit to compute the output SOH $\tilde{Q}(k)Q(k)$. The detailed process is as follows:

$$f(k) = \sigma(W_f[s(k-1), c(k)] + b_f) \tag{2}$$

$$i_k = \sigma(W_i[s(k-1), c(k)] + b_i) \tag{3}$$

$$\tilde{Q}_k = \tanh(W_c[s(k-1), c(k)] + b_c) \tag{4}$$

$$d(k) = f_k d(k-1) + i_k \tilde{C}_k \tag{5}$$

$$h(k) = \sigma(W_o[s(k-1), c(k)] + b_o) \tag{6}$$

In the equation, $W_f$, $W_i$, $W_c$, $W_o$ are weight matrices; $b_f$, $b_i$, $b_c$, $b_o$ are bias vectors; $s(k)$, $s(k-1)$ represent the hidden states of the $k$-th and $(k-1)$-th cycle, respectively; $c(k)$, $c(k-1)$ are the cell states of the current and previous modules, respectively; $\sigma$ is the sigmoid function.

### Feature optimization based on spatial attention mechanism

After short-term feature extraction, the optimal short-term features differ across different batteries. To ensure the model's generalizability, multiple features need to be selected. However, redundancy and interference exist among different features. In this section, a temporal attention mechanism is designed to eliminate redundancy between features. Therefore, a spatial attention mechanism is introduced to weight the sample features from $k, k-1, \cdots, k-M+1$, adaptively adjusting the short-term feature values. The model structure is shown in Fig 6. $x(k)$ represents the input features, composed of health factors F2 and F3, and after spatial attention weighting, new inputs $xs(k)$ are generated.

$$xs(k) = \sum_{i=1}^{T} \alpha^i(k) x_i(k), 1 \leq i \leq T \tag{7}$$

$$\alpha^i(k) = \frac{\exp|x_i(k) - Q(k-1)|}{\sum_{i=1}^{T} \exp|x_i(k) - Q(k-1)|}, 1 \leq i \leq T \tag{8}$$

The normalized value $\alpha^i(k)$ serves as the spatial attention weight, $x_i(k)$ is the three-dimensional vector composed of $F2(k)$, $F3(k)$, and $Q(k-1)$ is the actual capacity at the previous time step.

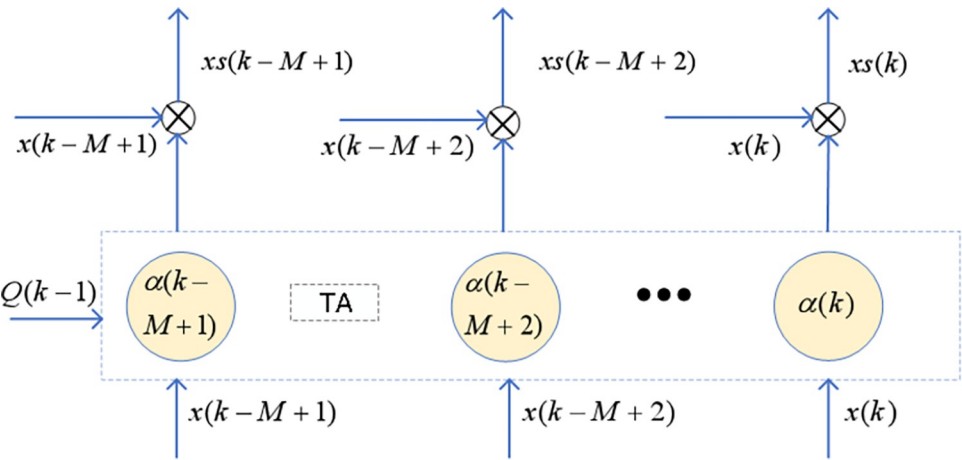

**Fig 6. Spatial attention feature weighting.**

## Feature optimization based on time attention mechanism

After the feature optimization by the spatial attention mechanism, to prevent abrupt feature changes caused by fluctuations in lithium-ion battery data, which could result in prediction outcomes severely deviating from normal values, a temporal attention mechanism is introduced. This mechanism applies weighting to the sample features at $k, k-1, \cdots, k-M+1$ to adaptively adjust the short-term feature values. As shown in Fig 7, the temporal attention value for the hidden state at time $k$ can be calculated as follows.

$$\beta_1^i(k) = \exp|xs_i(k) - s(k-1)|, 1 \leq i \leq M-1 \tag{9}$$

$$\beta_1^M(k) = \exp(|xs_M(k) - s(k-1)| + |xs_M(k) - xs_M(k-1)|) \tag{10}$$

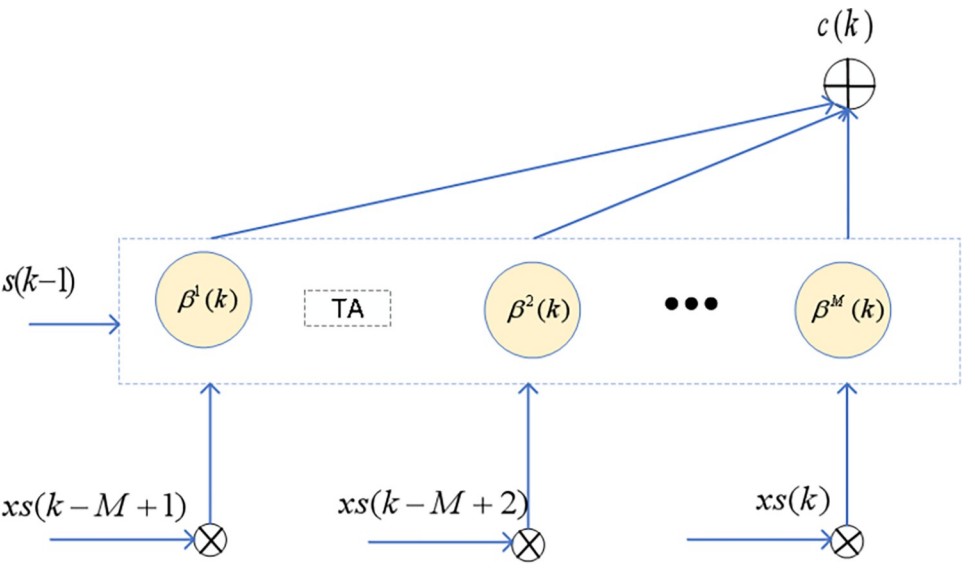

**Fig 7. Time attention features weighted.**

$$\beta^i(k) = \frac{\beta_1^i(k)}{\sum\limits_{i=1}^{M} \beta_1^i(k)}, 1 \le i \le M \tag{11}$$

$$c(k) = \sum_{i=1}^{M} \beta^i(k) x s_i(k), 1 \le i \le M \tag{12}$$

Where $\beta_1^i(k)$ is the weight of the i-th feature at time $k$, and $\beta_1^M(k)$ is the optimized weight of the M-th feature at time $k$. The normalized value $\beta^i(k)$ becomes the temporal attention weight, and $c(k)$ is the final feature input.

### Long-term trend feature extraction

To address the trend-like nonlinearity of the degradation sequence, which initially declines gradually and then rapidly, the degradation data is divided into trend data and short-term data. Trend data should reflect the characteristic of initially gradual and then rapid trends. Due to fluctuations caused by operational differences, the historical capacity exhibits significant variability, and a single capacity value cannot adequately express the trend pattern. Therefore, in this section, multiple average capacity values are selected as trend data $L(k)$:

$$L(k) = \frac{1}{M} \sum_{i=1}^{M} Q(k-i) \tag{13}$$

Where $Q(k-i)$ is the actual capacity. The LSTM model, with the short-term features $c(k)$, trend data $L(k)$, estimated capacity $\tilde{Q}(k-1)$, and actual capacity $Q(k-1)$ as inputs, is used to estimate the subsequent capacity $\tilde{Q}(k)$.

$$\tilde{Q}(k) = f_l[c(k), L(k), Q(k-1), \tilde{Q}(k-1)] + b_v \tag{14}$$

Where $f_l(\bullet)$ represents the LSTM unit, and $b_v$ is the bias vector.

### Algorithm description

The soft sensor model architecture based on STL-LSTM is shown in Fig 8, and the architecture is divided into three steps: (1) A complete lithium battery dataset is selected from the same

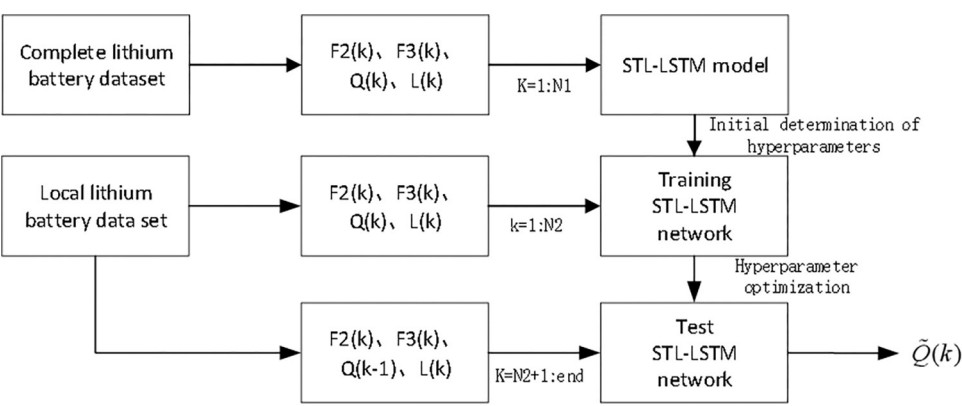

**Fig 8. Soft sensor modeling framework based on STL-LSTM.**

type of lithium batteries with a full SOH evolution dataset. Health factors $F2(k)$, $F3(k)$, the actual capacity $Q(k)$, and the trend feature $L(k)$ at time $k$ are extracted from this dataset. $k = 1,\ldots\ldots,N1$, where $k$ represents the cycle number, and N1 represents the number of cycles experienced when the capacity degrades to 70%. The STL-LSTM model is trained on this dataset to initially determine the network parameters; (2) The local lithium battery dataset refers to the battery currently in use. Health factors $F2(k)$, $F3(k)$, $Q(k)$ and $L(k)$ are extracted from its historical dataset. $k = 1,\ldots\ldots,N2$ where $k$ represents the cycle number, and N2 represents the selected cycle number used as training samples. The STL-LSTM model is trained on this data to optimize the network hyperparameters; (3) Based on $F2(k)$, $F3(k)$, $Q(k-1)$ and $L(k)$, the final capacity $\tilde{Q}(k)$ is estimated. The algorithm description of STL-LSTM is shown in Table 2.

## Experiment and analysis

### Data preparation

To validate the effectiveness of the proposed LSTM model based on the fusion of data characteristics and spatio-temporal attention mechanism (STL-LSTM) in enhancing the prediction

**Table 2. Description of STL-LSTM algorithm.**

| Algorithm: LSTM model based on data characteristics and space-time attention machine |
|---|
| Input: The input variable $x(k) = [x_1(k), x_2(k), \cdots, x_T(k)]$ represents the $k$-th cycle's short-term data, where T represents the dimensionality of the short-term data. $L(k)$ represents the real capacity value at the $k$-th cycle.<br>Output: The output variable $\tilde{Q}(k)$ represents the SOH estimated value for the $k$-th cycle's capacity. |
| 1.Algorithm: Long Short-Term Memory Network Based on Temporal Attention Mechanism(STL-LSTM) procedure:<br>1.Standardize the data set<br>2.Initialize the network parameters, including the weight matrix W and the bias vectors U,V<br>3.Execute for each time step $T = 1$ to $k$:<br>a.Short-time feature extraction:<br>For each time step $k = 1$ to $T$<br>i. Calculate the activation values of the input, forget, and output gates<br>Input gate: $i_k = \sigma(W_i[s(k-1), c(k)] + b_i)$<br>Forget Gate: $f(k) = \sigma(W_f[s(k-1), c(k)] + b_f)$<br>Output gate: $h(k) = \sigma(W_o[s(k-1), c(k)] + b_o)$<br>Calculate cell state<br>$\tilde{Q}_k = \tanh(W_c[s(k-1), c(k)] + b_c)$<br>Update the hidden status<br>$d(k) = f_k d(k-1) + i_k \tilde{C}_k$<br>b. Spatial attention mechanism:<br>For each time step $k = 1$ to T:<br><br>i.Calculate the time attention value at moment $k$: $xs(k) = \sum_{i=1}^{T} \alpha^i(k) x_i(k), 1 \leq i \leq T$<br><br>ii.The time weighted sum of all encoder hidden states is calculated to obtain the final feature input:<br>$\alpha^i(k) = \dfrac{\exp|x_i(k) - Q(k-1)|}{\sum_{i=1}^{T} \exp|x_i(k) - Q(k-1)|}, 1 \leq i \leq T$<br>b. Temporal attention mechanism:<br>i.Calculate the time attention value of the hidden state at time $k$:<br>$\beta_1^i(k) = \exp|xs_i(k) - s(k-1)|, 1 \leq i \leq M-1$<br>ii.Final feature input $c(k)$<br>$c(k) = \sum_{i=1}^{M} \beta^i(k) xs_i(k), 1 \leq i \leq M$<br>c. Capacity estimation:<br>i.The average capacity is trend data $L(k)$<br>$L(k) = \frac{1}{M} \sum_{i=1}^{M} Q(k-i)$<br>ii. $c(k), L(k), \tilde{Q}(k-1), Q(k-1)$ as input of LSTM model as input, Estimate subsequent capacity $\tilde{Q}(k)$<br>$\tilde{Q}(k) = f_l[c(k), L(k), Q(k-1), \tilde{Q}(k-1)] + b_v$ |

accuracy of State of Health (SOH) for lithium-ion batteries, data from the NASA battery database was used. Within the dataset, battery models and time points $F2(k)$, $F3(k)$, $Q(k-1)$ and $L(k)$ were selected to estimate capacity $\tilde{Q}(k)$. Specifically, batteries B5 to B7 contributed data for 168 cycles each, and the complete datasets from batteries B5 and B6 were utilized as focal study datasets for model training. The STL-LSTM model was trained on SOH databases indicating capacity degradation from 50% to 70%, showing a relatively higher error rate when predictions did not reach the anticipated range [30]. To evaluate the performance of the algorithm, Root Mean Square Error (RMSE) and Mean Absolute Error (MAE) were used as evaluation metrics, defined as follows:

$$RMSE = \sqrt{\frac{1}{n}\sum_{i=1}^{n}\left(\hat{Q}(k) - Q(k)\right)^2} \qquad (15)$$

$$MAE = \frac{1}{n}\sum_{i=1}^{n}|\hat{Q}(k) - Q(k)| \qquad (16)$$

Where $\hat{Q}(k)$ is the predicted value for the $k$-th cycle, and $Q(k)$ is the actual value for the $k$-th cycle.

## Algorithm performance comparison under different characteristics

To verify the impact of different features on the capacity estimation performance of the LSTM model, this section selects three features that are highly correlated with the capacity at time $k$: the capacity $Q(k-1)$ at time $k-1$, and the health factors $F2(k)$, $F3(k)$. Three models were designed accordingly: the first method uses $Q(k-1)$ as the input feature, the second method uses $F2(k)$, $F3(k)$ as the input features, and the third method uses $Q(k-1)$, $F2(k)$, $F3(k)$ as input features. For the first method, $Q(k-1)$ is highly correlated with $Q(k)$ when the battery capacity changes slowly. However, when the battery has not been used for an extended period, a capacity regeneration phenomenon occurs, and there is a significant difference between $Q(k-1)$ and $Q(k)$, manifested as sharp peaks in the battery capacity curve. For the second method, since the health factors $F2(k)$, $F3(k)$ are highly related to the charged capacity at time k, a large charged capacity necessarily means a large discharged capacity. Therefore, theoretically, using $F2(k)$, $F3(k)$ as inputs should result in better performance, especially in estimating the sharp peaks during capacity regeneration. For the third method, which selects three features, theoretically, $Q(k-1)$ should benefit capacity estimation performance in regions of slow capacity change, while $F2(k)$, $F3(k)$ should improve peak capacity estimation.

To verify and compare the performance of the three methods, this section selects batteries B5 and B6 from the NASA dataset, with the first 50% and 70% of the battery data used as the training set, and the remaining 50% and 30% as the test set. The experimental results are shown in Figs 9 and 10, Table 3. From the figures and table, the following can be observed:

1)Methods 1 and 2 provide estimated capacities that are close to the actual capacity when the battery capacity is in a region of slow change. However, when the capacity is in a sharp peak region, Method 1 shows a larger estimation error compared to the actual capacity, whereas Method 2 has a significantly smaller error, which is consistent with the theoretical analysis. Method 3 performs worse than Methods 1 and 2 in regions of slow capacity change due to the interference among the simple combinations of the three features, which negatively impacts the estimation accuracy.

2)When using the first 50% and 70% as the training set, the performance metrics of the three methods ranked from best to worst are: Method 2, Method 1, and Method 3. However,

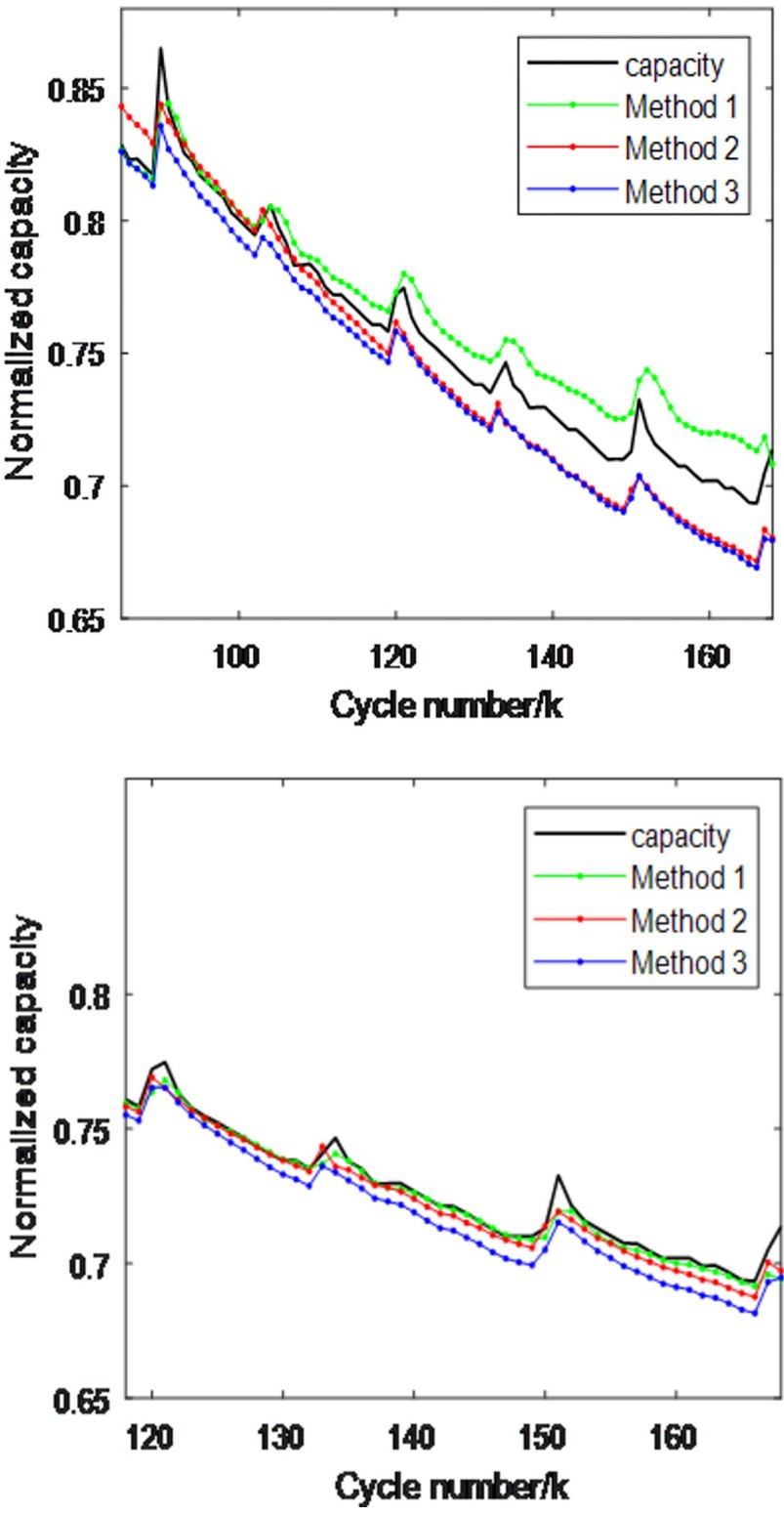

**Fig 9. B5 battery capacity estimation performance comparison of three methods.** (a) The first 50% is the training set (b)The first 70% is the training set.

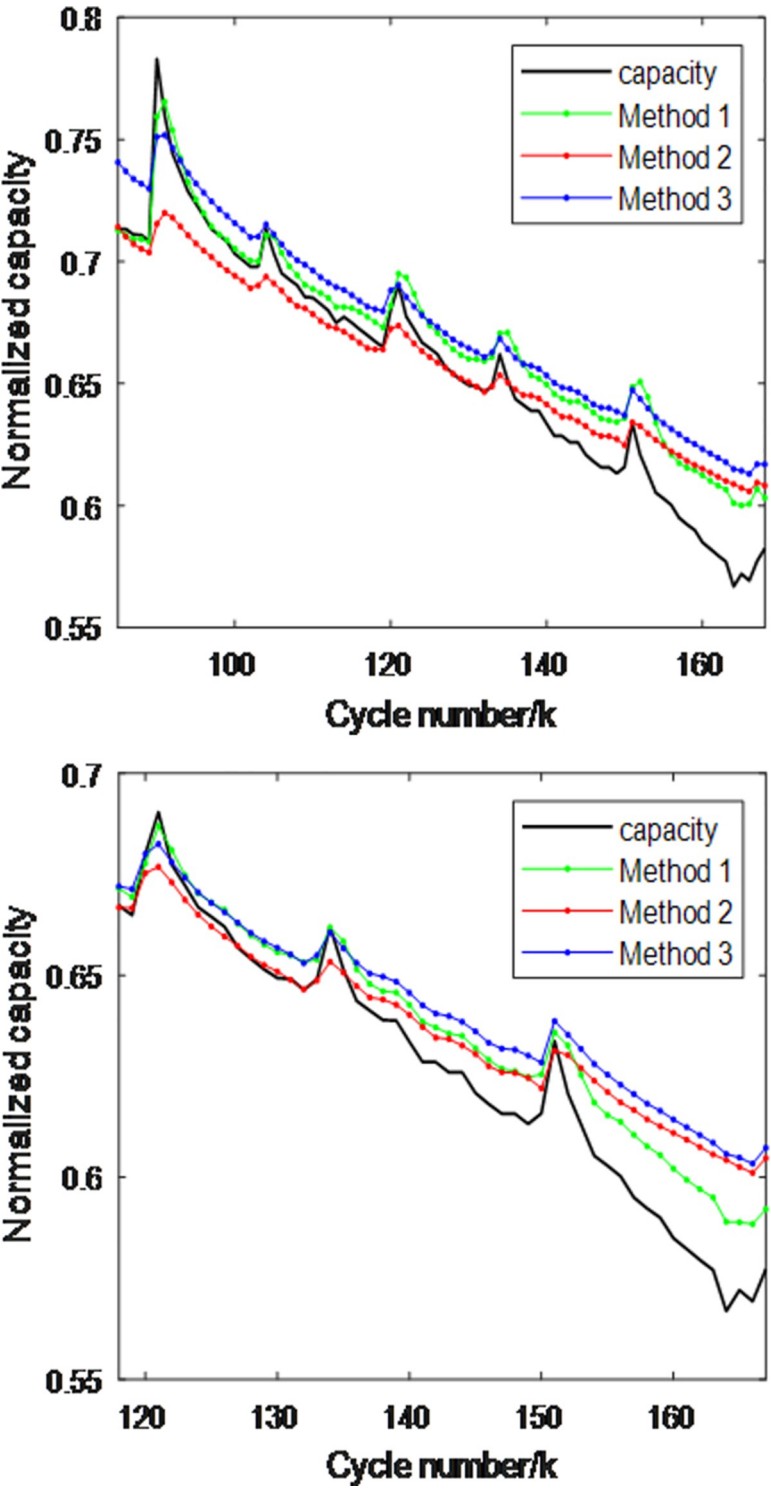

**Fig 10. B6 battery capacity estimation performance comparison of three methods.** (a)The first 50% is the training set (b)The first 70% is the training set.

**Table 3. Comparison of capacity estimation performance metrics for two batteries using three methods.**

| Battery | Training Set | Method 1<br>RMSE/MAE<br>$(10^{-2})$ | Method 2<br>RMSE/MAE<br>$(10^{-2})$ | Method 3<br>RMSE/MAE<br>$(10^{-2})$ |
|---|---|---|---|---|
| B0005 | 50% | 1.49/1.26 | 1.45/1.24 | 1.60/1.46 |
| B0005 | 70% | 0.59/0.36 | 0.48/0.36 | 0.89/0.83 |
| B0006 | 50% | 2.00/1.60 | 1.76/1.30 | 2.15/1.86 |
| B0006 | 70% | 1.34/1.25 | 1.50/1.10 | 1.78/1.46 |

with an increase in the training set size, the performance metrics of all three methods significantly improve, indicating that expanding the training set enhances estimation performance.

## Impact of different modules on algorithm performance

To verify the impact of spatio-temporal attention and long-term trend features on algorithm performance, this paper designs three methods: the SOH estimation model based on spatial attention mechanism (SA-LSTM), the SOH estimation model based on spatio-temporal attention mechanism (ST-LSTM), and the method proposed in this paper, STL-LSTM. In the SA model, the temporal attention mechanism module is removed from the original STL-LSTM model, while in the ST-LSTM model, the long-term trend feature input is removed [50]. To verify and compare the performance of the three methods, the NASA dataset with batteries B5 and B6 was selected, with the first 50% and 70% of the battery data used as the training set and the remaining 50% and 30% as the test set. The experimental results are shown in Figs 11 and 12, Table 4. From the figures and table, the following can be observed:

1. In the capacity regeneration region, the SA-LSTM model performs best at time $k+1$. This is because the SA at time $k+1$ integrates the health factors F2 and F3, which are highly correlated with the capacity $Q$, making it beneficial for estimating $Q$. However, this also exacerbates dynamic fluctuations, leading to greater estimation performance variation after model training. At time $k+2$, the performance of STL-LSTM is close to that of SA-LSTM because the temporal attention mechanism introduces adaptive weighting, increasing the fusion weight of F2 and F3 at the most recent time when the F2 and F3 values show a trend of increase or decrease.

2. In the early slow-changing region, the performance of ST-LSTM is significantly better than that of SA-LSTM, and the performance of ST-LSTM is close to that of STL-LSTM.

3. In the later stages, the overall performance of STL-LSTM is closer to the actual capacity and is superior to STL-LSTM. This is because the overall trend feature was introduced, ensuring that the model can achieve both short-term and long-term fusion estimation.

## Performance comparison of different algorithms

To compare the superiority of the STL-LSTM algorithm proposed in this paper, the mainstream LSTM algorithm was selected for comparison, using the NASA datasets for batteries B5 and B6. The first 70% of the battery data was used as the training set, and the remaining 30% as the test set. The experimental results are shown in Table 5. From the table, it can be seen that the STL-LSTM algorithm outperforms the traditional algorithm in terms of performance in short-term regions, regeneration regions, and later-stage regions.

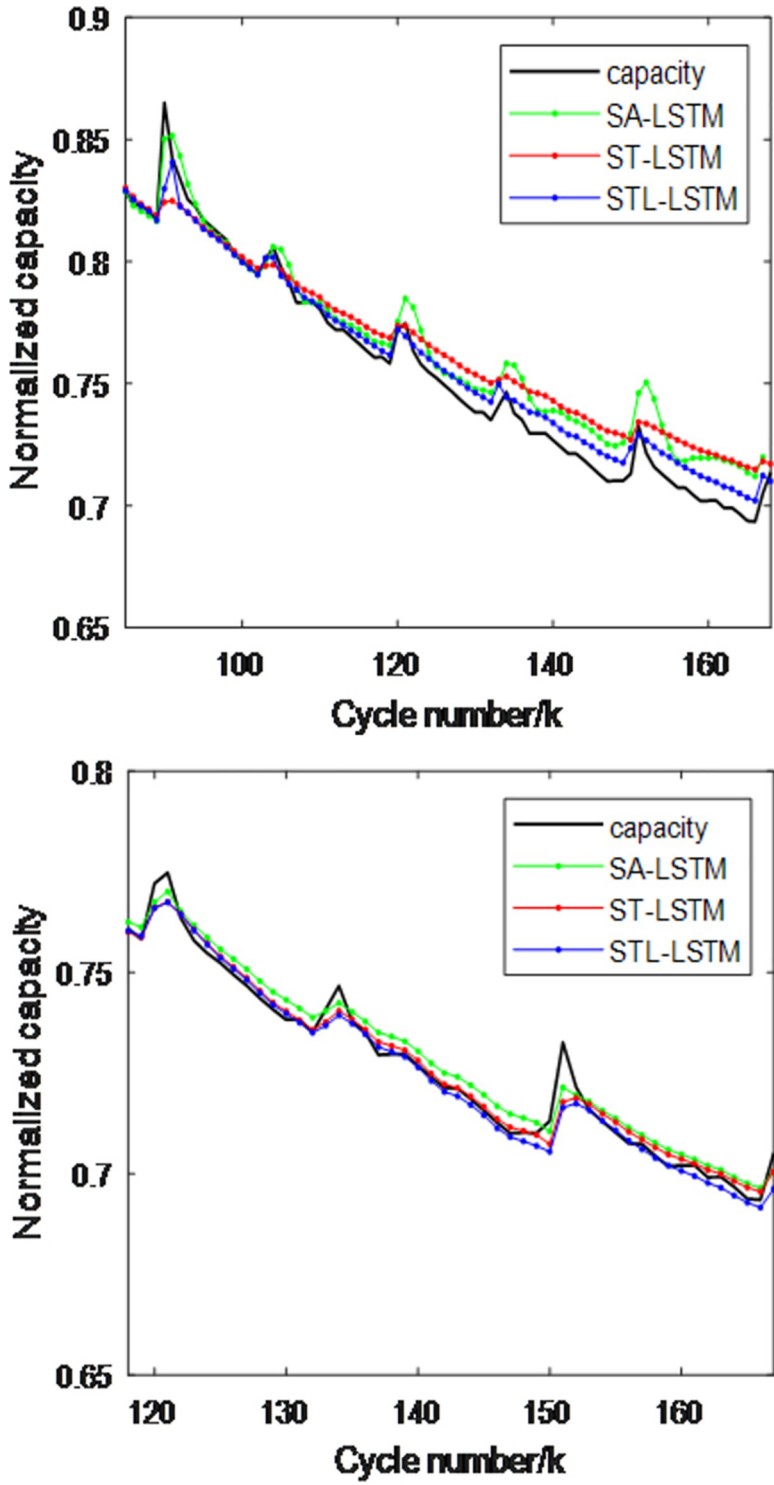

**Fig 11. Comparison of capacity estimation performance for battery B5 using three methods.** (a)The first 50% is the training set (b)The first 70% is the training set.

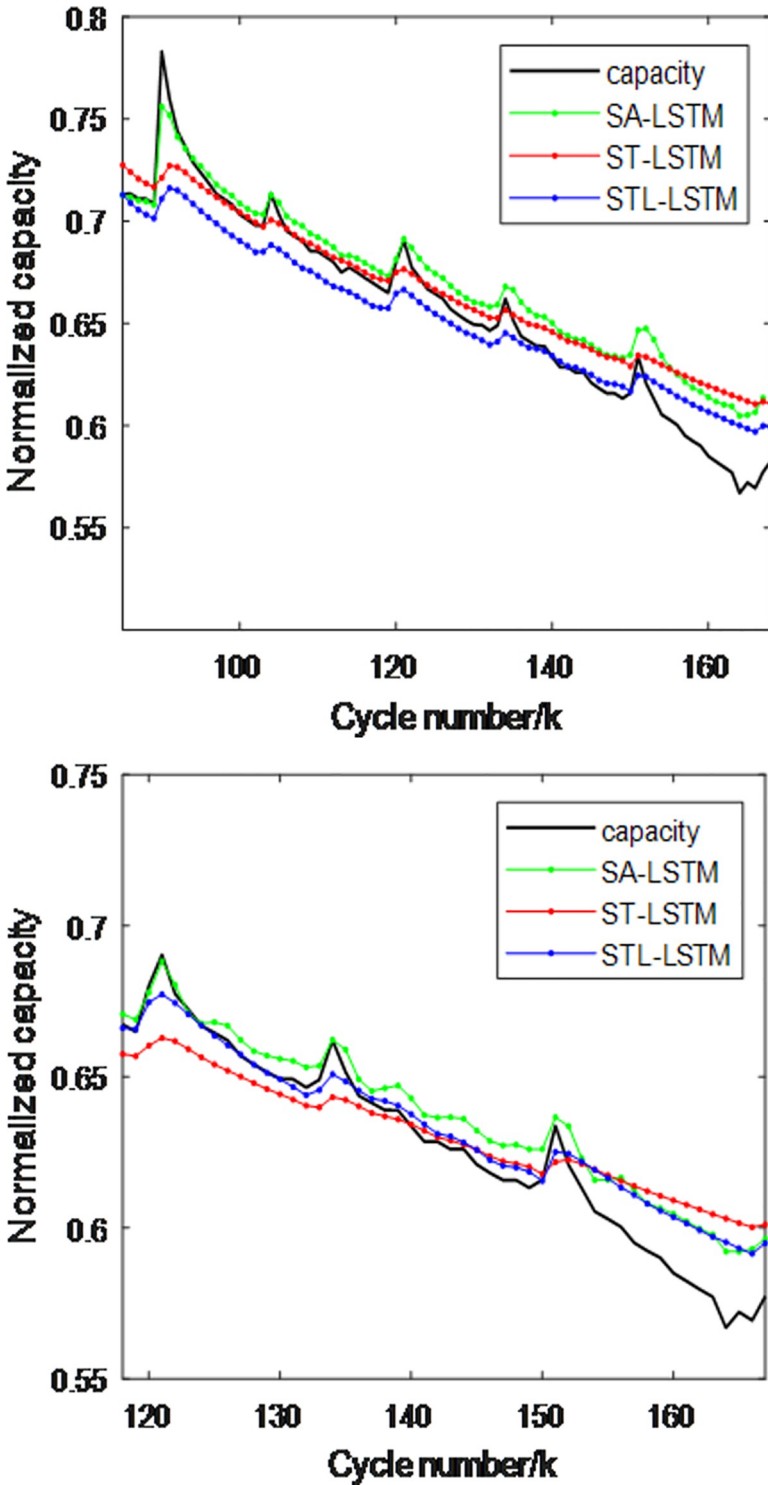

**Fig 12. Comparison of capacity estimation performance for battery B6 using three methods.** (a)The first 50% is the training set (b)The first 70% is the training set.

Table 4. Comparison of capacity estimation performance metrics for two batteries using three attention mechanism models.

| Battery | Training Set | SA-LSTM RMSE/MAE $(10^{-2})$ | ST-LSTM RMSE/MAE $(10^{-2})$ | STL-LSTM RMSE/MAE $(10^{-2})$ |
|---|---|---|---|---|
| B0005 | 50% | 1.47/1.20 | 1.35/1.13 | 0.75/0.59 |
| B0005 | 70% | 0.51/0.32 | 0.42/0.37 | 0.38/0.24 |
| B0006 | 50% | 1.42/1.29 | 1.51/1.21 | 1.06/0.76 |
| B0006 | 70% | 2.01/1.63 | 1.81/1.28 | 1.66/1.19 |

Table 5. Comparison of different LSTM algorithms.

| Comparison of methods | RMSE($10^{-2}$) | MSE($10^{-4}$) | MAE($10^{-2}$) | MAPE |
|---|---|---|---|---|
| CNN-LSTM | 0.46 | 0.29 | 0.36 | 0.31 |
| Bi-LSTM | 0.41 | 0.27 | 0.40 | 0.28 |
| STL-LSTM | 0.38 | 0.25 | 0.24 | 0.20 |

## Conclusion

Conventional LSTM-based SOH estimation methods do not account for the trend-like nonlinearity in battery SOH degradation sequences and the significant dynamic variations between samples. This paper proposes an LSTM-based lithium battery SOH estimation method that incorporates data characteristics and spatio-temporal attention. Considering the trend-like nonlinearity of degradation sequences, which starts gradually and then accelerates, the input features are divided into trend and non-trend features. The trend data reflects the gradual-to-rapid change in the trend, while the non-trend features include health factors and the actual capacity at the previous time step. To address the significant dynamic variations between samples, especially the issue of capacity regeneration, a spatio-temporal attention mechanism structure for multidimensional non-trend features is designed to extract spatio-temporal features. Subsequently, an LSTM model is built using trend features, spatio-temporal features, actual capacity, and other inputs to estimate the capacity. Finally, the model is trained and tested on different datasets. Experimental results demonstrate that STL-LSTM outperforms traditional algorithms in short-term regions, regeneration regions, and later-stage regions.

## Author Contributions

**Conceptualization:** Jingyun Xu.

**Data curation:** Yifan Zhu.

**Formal analysis:** Jingyun Xu.

**Software:** Gengchen Xu.

**Writing – original draft:** Gengchen Xu.

**Writing – review & editing:** Gengchen Xu.

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
