## [Decision Letter · Decision Letter 0]

2 Sep 2024

PONE-D-24-34452LSTM-Based Estimation of Lithium-Ion Battery SOH Using Data Characteristics and Spatio-Temporal AttentionPLOS ONE

Dear Dr. xu,

Thank you for submitting your manuscript to PLOS ONE. After careful consideration, we feel that it has merit but does not fully meet PLOS ONE’s publication criteria as it currently stands. Therefore, we invite you to submit a revised version of the manuscript that addresses the points raised during the review process.

The manuscript presents a LSTM-based estimation of lithium-ion battery SOH using data characteristics and spatio-temporal attention. Generally, it is well prepared. But there are still some place that need further revisions or clarifications before considering for this joural publication.

We look forward to receiving your revised manuscript.

Kind regards,

Lei Zhang, PhD

Academic Editor

PLOS ONE

Journal Requirements:

"This research was partially supported by Zhejiang Provincial Natural Science Foundation of China under Grant (No. LTGS23E070002)."

 “The authors received no specific funding for this work.”

"There are no competing interests"

6. PLOS requires an ORCID iD for the corresponding author in Editorial Manager on papers submitted after December 6th, 2016. Please ensure that you have an ORCID iD and that it is validated in Editorial Manager. To do this, go to ‘Update my Information’ (in the upper left-hand corner of the main menu), and click on the Fetch/Validate link next to the ORCID field. This will take you to the ORCID site and allow you to create a new iD or authenticate a pre-existing iD in Editorial Manager.

Additional Editor Comments:

The manuscript presents a LSTM-based estimation of lithium-ion battery SOH using data characteristics and spatio-temporal attention. Generally, it is well prepared. But there are still some place that need further revisions or clarifications before considering for this joural publication.

Reviewers' comments:

Reviewer's Responses to Questions

**Comments to the Author**

1. Is the manuscript technically sound, and do the data support the conclusions?

Reviewer #1: Yes

Reviewer #2: Yes

2. Has the statistical analysis been performed appropriately and rigorously? 

Reviewer #1: Yes

Reviewer #2: Yes

3. Have the authors made all data underlying the findings in their manuscript fully available?

Reviewer #1: Yes

Reviewer #2: Yes

4. Is the manuscript presented in an intelligible fashion and written in standard English?

Reviewer #1: No

Reviewer #2: Yes

5. Review Comments to the Author

Reviewer #1: This paper tries to propose a novel battery SOH estimation strategy based on combination of LSTM model and Spatio-Temporal Attention method. However, the authors fail to clearly demonstrate how the proposed strategy working and why it is better than others. Besides, there are lots of grammar, writing and organizing deficiencies of this manuscript. It is difficult to recommend publication in this version of paper. The detailed comments are list blow:

1. The writing and organizing of this manuscript are far from satisfactory and professional. For example:

a) The name of authors in the information table.

b) The figure quality in this manuscript is less appreciate, including font, size, capitalize, and unit.

c) The table in this paper is also low quality, including font, header name, organization, key information highlight, and so on.

d) What is the meaning of ‘L’ in the vector parameters?

e) There are serious formatting issues with the formula and the related explanation.

f) Inconsistent capitalization in Section titles.

2. The NASA datasets have been widely used in numerous articles and are outdated. The authors should examine the proposed method based on other recent datasets with different battery types.

3. Four health factors are very important in this paper but the authors ignore to describe them clearly. Meanwhile, these factors are all related with charging time, which is possible to be redundant.

4. The conclusion of correlation analysis is unclear. Which factor should be selected and used? Why the authors emphasize the correlation between capacity and factors? In fact, there are lots of discussion without conclusive result.

5. How to divide the degradation data into trend data and short-term data? What is the definition of the trend and short-term data?

6. It is hard to fully understand the Fig.4 without explicit explanation.

7. Section Ⅲ is the core of this paper but describe confusedly. Thus, the readers can hardly understand the contribution and innovation of this work.

8. The authors could investigate the public paper for better writing and organizing. For example:

a) Battery SOH estimation method based on gradual decreasing current, double correlation analysis and GRU

b) Performance simulation method and state of health estimation for lithium-ion batteries based on aging-effect coupling model

c) Accurate state of health estimation of battery system based on multi-stage constant current charging and behavior analysis in real-world electric vehicles

Reviewer #2: This paper introduces an advanced lithium-ion battery SOH estimation method using LSTM network, which combines trend and non-trend features with a spatiotemporal attention mechanism. The method effectively addresses trend-based nonlinearities and significant dynamic changes in SOH degradation sequences. However, there are still several aspects of the current manuscript that require improvement before it can be considered for publication.

1. In section 2, the five different discharge rates and intervals were not visualized, it is recommended to add some details about it.

2. The reviewer wants to know why the author choose B5,B6,B7 among the NASA batteries datasets as the subjects for health factor extraction.

3. On page 7, line 11, “For the batteries B5, B7, the F1 health factor shows the strongest correlation with the capacity degradation value.” it is illustrated in table1 that it is the health factor F4 of B7 shows the strongest correlation.

4. The reviewer would like to know that why F4 health factor of B6 shows a big difference comparing with other factors.

5. The reviewer wonders that what are the specific meanings and advantages of the performance evaluation metrics (such as RMSE) the author chose respectively in this experiment.

6. On page 17, line 22, “Using features from the current cycle, the model has a good ability to generalize to unseen data.” there are not adequate descriptions that supporting this view, it is recommended to add some further explanations.

7. For battery health estimation, it is recommended to include the following studies to enhance the literature review. “Rapid health estimation of in-service battery packs based on limited labels and domain adaptation” and “Data-Driven Battery State of Health Estimation Based on Random Partial Charging Data”.

8. Others

On page 5, the word RUL is not predefined.

6. PLOS authors have the option to publish the peer review history of their article (what does this mean?). If published, this will include your full peer review and any attached files.

Reviewer #1: No

Reviewer #2: No

---

## [Author Response · Author response to Decision Letter 0]

27 Sep 2024

I have provided detailed responses to the reviewers' comments, which have been submitted as an attachment. Thank you for reviewing my work and for your valuable suggestions.

---

## [Decision Letter · Decision Letter 1]

15 Oct 2024

LSTM-Based Estimation of Lithium-Ion Battery SOH Using Data Characteristics and Spatio-Temporal Attention

PONE-D-24-34452R1

Dear Dr. Xu,

We’re pleased to inform you that your manuscript has been judged scientifically suitable for publication and will be formally accepted for publication once it meets all outstanding technical requirements.

Kind regards,

Lei Zhang, PhD

Academic Editor

PLOS ONE

Additional Editor Comments (optional):

The revised manuscript has addressed all the raised questions, and thus qualifies for this journal publication.

Reviewers' comments:

Reviewer's Responses to Questions

**Comments to the Author**

1. If the authors have adequately addressed your comments raised in a previous round of review and you feel that this manuscript is now acceptable for publication, you may indicate that here to bypass the “Comments to the Author” section, enter your conflict of interest statement in the “Confidential to Editor” section, and submit your "Accept" recommendation.

Reviewer #2: All comments have been addressed

Reviewer #3: All comments have been addressed

2. Is the manuscript technically sound, and do the data support the conclusions?

Reviewer #2: Yes

Reviewer #3: Yes

3. Has the statistical analysis been performed appropriately and rigorously? 

Reviewer #2: Yes

Reviewer #3: Yes

4. Have the authors made all data underlying the findings in their manuscript fully available?

Reviewer #2: No

Reviewer #3: Yes

5. Is the manuscript presented in an intelligible fashion and written in standard English?

Reviewer #2: Yes

Reviewer #3: Yes

6. Review Comments to the Author

Reviewer #2: After careful revision by the authors, the quality of the paper has been improved. I have no more questions.

Reviewer #3: The authors have answered all of questions. I have no further questions. And the paper is in good form, can be accepted.

7. PLOS authors have the option to publish the peer review history of their article (what does this mean?). If published, this will include your full peer review and any attached files.

Reviewer #2: No

Reviewer #3: No

---

## [Editor Report · Acceptance letter]

18 Oct 2024

PONE-D-24-34452R1 

PLOS ONE

Dear Dr. Xu, 

I'm pleased to inform you that your manuscript has been deemed suitable for publication in PLOS ONE. Congratulations! Your manuscript is now being handed over to our production team.

Kind regards, 

on behalf of

Dr. Lei Zhang 

Academic Editor

PLOS ONE